# Biomarkers and Biochemical Indicators to Evaluate Bone Metabolism in Preterm Neonates

**DOI:** 10.3390/biomedicines12061271

**Published:** 2024-06-07

**Authors:** Gabriele D’Amato, Vincenzo Brescia, Antonietta Fontana, Maria Pia Natale, Roberto Lovero, Lucia Varraso, Francesca Di Serio, Simonetta Simonetti, Paola Muggeo, Maria Felicia Faienza

**Affiliations:** 1Neonatal Intensive Care Unit, Di Venere Hospital, 70012 Bari, Italy; gab59it@yahoo.it (G.D.); marina.n83@libero.it (M.P.N.); 2Clinical Pathology Unit, AOU Policlinico Consorziale di Bari-Ospedale Giovanni XXIII, 70124 Bari, Italy; antonietta.fontana@policlinico.ba.it (A.F.); roberto.lovero@policlinico.ba.it (R.L.); lucia.varraso@policlinico.ba.it (L.V.); francesca.diserio@policlinico.ba.it (F.D.S.); 3Clinical Pathology and Neonatal Screening, AOU Policlinico Consorziale di Bari-Ospedale Giovanni XXIII, 70124 Bari, Italy; simonetta.simonetti@policlinico.ba.it; 4Department of Pediatric Oncology and Hematology, AOU Policlinico Consorziale di Bari-Ospedale Giovanni XXIII, 70124 Bari, Italy; paola.muggeo@gmail.com; 5Pediatric Unit, Department of Precision and Regenerative Medicine and Ionian Area, University of Bari “A. Moro”, 70124 Bari, Italy; mariafelicia.faienza@uniba.it

**Keywords:** preterm newborns, bone biomarkers, total alkaline phosphatase (ALP), collagen type I amino-terminal propeptide (PINP), osteocalcin, collagen type 1 carboxyl-terminal telopeptide (CTX), leptin, metabolic bone disease (MBD) of prematurity

## Abstract

The purpose of the present study was to evaluate the concentrations of some bone turnover markers in preterm neonates with uncomplicated clinical course in the first month of life. Samples from 13 preterm neonates were collected at three different times: at birth (T0) from umbilical cord blood (UCB); and at 15 (T1) and 30 (T2) days of life from peripheral blood (PB). The concentrations of calcium (Ca), phosphate (P), total alkaline phosphatase (ALP), Collagen Type 1 Amino-terminal Propeptide (PINP), osteocalcin (OC), Collagen Type 1 Carboxyl-Terminal Telopeptide (CTX) and Leptin were assessed. A statistically significant difference for ALP concentration at birth versus T1 and T2 was found. An evident increase in the median concentrations of CTX, OC and PINP from T0 to T2 were observed. A significant difference was also found for Leptin concentration at T0 compared to T1. In preterm infants, in the absence of acute or chronic medical conditions and without risk factors for metabolic bone disease (MBD) of prematurity, there is a significant increase in bone turnover markers during the first month of life. The knowledge of the variations in these markers in the first weeks of life, integrated by the variations in the biochemical indicators of bone metabolism, could help in recognizing any conditions at risk of developing bone diseases.

## 1. Introduction

During fetal growth, the placenta actively transports calcium, magnesium and phosphate from the maternal circulation, especially in the third trimester of pregnancy when 80% of mineral content is achieved [1]. Bone mineralization is a process that begins in matrix vesicles derived from both chondrocytes and osteoblasts, considered the bone-forming cells. Calcium (Ca) and phosphate ions (P), through their specific channels, are captured by the matrix vesicles and crystallize to form hydroxyapatite; subsequently, the hydroxyapatite spreads on the collagen fibrils and mineralizes the extracellular matrix [2]. A higher extracellular concentration of minerals compared to the maternal circulation allows the fetus to adequately mineralize the skeleton before birth. Indeed, the availability of minerals influences the activity of osteoblasts, the bone-forming cells, and osteoclasts, the bone-reabsorbing cells [3].

Furthermore, during pregnancy and at birth, elevated serum Leptin levels have been found, probably due to placental production. Leptin is a hormone predominantly produced by adipose cells, and its main function is likely to be to regulate long-term energy balance [4]. It has a central action on the hypothalamus, while non-hypothalamic targets of Leptin are represented by several organs and tissues such as the fetal lung, circulatory system, reproductive system, immunological system and bone. Its effects on bone are mediated directly by signals (neuropeptides and neurotransmitters) from the brain. Leptin action in decreasing cancellous bone and increasing the cortical bone formation could represent a mechanism for augmenting bone size and resistance [5]. Maternal and fetal serum Leptin levels are dysregulated in pathological conditions, such as gestational diabetes, preeclampsia and intra-uterine growth retardation (IUGR), as a consequence of a disruption in the fetal/placental/maternal unit. Moreover, low Leptin levels have been found in the umbilical cord blood of preterm neonates, although they would not appear to affect bone mineral density later in life [6].

Other biomarkers have also been studied in preterm infants, in particular Type 1 Amino-Terminal Propeptide (PINP), osteocalcin (OC) and Collagen Type 1 Car-boxyl-Terminal Telopeptide (CTX) [7].

PINP is a marker of bone formation; it is a 35KD protein produced by the cleavage of type 1 procollagen, synthesized by osteoblasts, and the cleavage occurs by specific endopeptidases that release PINP. PINP was related to leg growth velocity in very low birth weight infants [7].

Osteocalcin is a 49 kDa protein secreted by osteoblasts into the bone matrix. OC is generally considered an indicator of non-collagenous bone formation. In particular, OC appears to be a very specific predictor of reduced bone mass in premature infants with MBD [4].

CTX derives from the C-terminal portion of type I collagen and is one of the main biomarkers of bone resorption [7].

Preterm neonates are born in a period of gestation in which the highest mineral accretion rate of calcium (120–200 mg/kg/day) is required [8]. This requirement is not satisfied in extrauterine life, and therefore the mineralization of the skeleton is compromised to maintain the mineral concentration in the serum. This condition can be further impaired by Ca absorption from the immature intestine. In addition, poor oral Ca and P intake further impairs mineral homeostasis, resulting in the development of metabolic bone disease (MBD) of prematurity in high-risk preterm infants [9,10].

Metabolic bone disease (MBD) of prematurity is a bone health disorder characterized by hypophosphatemia, increased total ALP and late onset of radiological findings of bone demineralization [11,12]. Clinical signs appear between 5 and 11 weeks of life and are characterized by a widening of the cranial sutures, frontal protuberances, rickets, fractures, increased work of breathing due to instability of the chest wall caused by softening or fractures of the ribs and postnatal growth retardation [13,14].

Gestational age (GA), birth weight, use of glucocorticoids in infants with chronic lung disease (CLD), infections and total prolonged parenteral nutrition (PN) with a delay in establishing and progressing enteral nutrition represent risk factors for an altered bone formation [14]. In low birth weight infants, there is a great variability in Ca retention (80–170 mg/kg/day) possibly due to a significant individual susceptibility to Ca metabolism [15,16]. With gut maturity advancing, in later postnatal GA, there is an efficient retention of Ca due to a change towards an active, calcitriol-dependent, saturable mechanism of absorption [17]. There are no specific diagnostic methods for MBD of prematurity in individuals who are at risk. Clinical results appear late and sometimes the diagnosis is not made [10].

Increased bone turnover has been previously reported in preterm infants, especially those with MBD. Several authors have highlighted that an approach to the diagnosis of MBD based on a combination of different biomarkers of bone formation and resorption appears to be more useful. In fact, the indices of bone formation osteocalcin (OC), Type 1 Amino-Terminal Propeptide (PINP) and resorption (Collagen Type 1 Carboxyl-Terminal Telopeptide (CTX) are significantly increased in the group of newborns with MBD, both term and premature [4].

For the limitation of single predictors of MBD, an approach to the diagnosis of MBD based on a combination of several factors, including markers of bone formation and reabsorption, appears to be more encouraging [18].

In the present study, we aimed to determine the serum levels of some bone turnover markers in preterm neonates, to evaluate the trend in their concentrations in the first month of life after placental separation. For this purpose, preterm neonates with uncomplicated clinical course and who were not predisposed to MBD were selected to reduce, the interferent factors with bone metabolism. Biomarkers of bone formation, such as N-terminal propeptide of type-1 procollagen (PINP) and osteocalcin (OC), and a marker of bone resorption, such as C-terminal telopeptide of type I collagen (CTX-I), were evaluated in conjunction with serum Ca, P, total alkaline phosphatase (ALP) concentrations and with Leptin, with the purpose of following the bone formation and remodeling process after birth.

## 2. Materials and Methods 

### 2.1. Subjects

For this study, we included 13 Caucasian preterm neonates (7 males) with gestational age (GA) ranging from 28 to 35 weeks, born to primiparous mothers 9 (64%), with an average age of 29 years (range 25–37) with a medium–high level of education. Based on information based on pre-pregnancy BMI, 10 (77%) women were of normal weight BMI between 18.5 and 24.9 kg/m^2^ and 3 (23%) were overweight with a BMI between 25.0 and 29.9 kg/m^2^ according to the World Health Organization classification (BMI classification 2006). The anthropometric measurements of the neonates were determined at birth. Thereafter, all infants were weighed daily, and their length and the head circumference measured weekly. The body weight was measured with an accuracy of ±5 g. The length and head circumference were measured to the nearest centimeter and millimeter. The values of length, weight and head circumference were plotted on the INeS Charts, and the Z-score was evaluated. The exclusion criteria for enrollment and statistical analysis were acute or chronic illness such as moderate/severe respiratory distress syndrome (RDS), sepsis, CLD requiring treatment with glucocorticoids and/or diuretics, cholestatic jaundice due to prolonged PN; necrotizing enterocolitis and feeding intolerance with abdominal distension predicting prolonged suspension of enteral nutrition; congenital malformations or genetic syndromes; secondary hyperparathyroidism with or without hypophosphatemia and hypocalcemia.

The following observational study was approved by the ethical committees of the Bari University Hospital (Biomarkers of bone metabolism; Protocol number 5849/PRETERM). This study was conducted in accordance with the principles of the Declaration of Helsinki and the “International Conference on Harmonization Guidelines for Good Clinical Practice”. Informed parental consent was obtained.

### 2.2. Samples Collection and Storage

The blood for biochemical markers and biomarkers of bone metabolism were drawn from the umbilical cord (mixed venous and arterial blood) (time 0; T0) and from the peripheral vein of neonates after 15 days (time 1; T1) and 30 days of life (time 2; T2) for a total of 39 samples. The blood samples were centrifuged at 3000 rpm for 10 min, in order to physically separate the serum from the cellular component of the blood. Approximately 1.5 mL of serum was collected for each sample. The serum was frozen at −80 °C and thawed at room temperature before being tested. All samples were analyzed in a single session, avoiding repeated freeze–thaw cycles.

### 2.3. Serum Biochemical Markers

Total calcium (Ca) and phosphate (P) were assayed by the spectrophotometrics method and total alkaline phosphatase (ALP) was carried out by the enzyme method (bichromatic kinetic technique) using the Dimension VISTA 1500 instrument (Siemens, Munich, Germany). These are colorimetric analyses based on the principle whereby the concentration of the substance to be measured will be proportional to the color variations that occur in the specific reaction, when the substance is exposed to a beam of light of intensity (I0). The reference values for Ca, P and ALP were 2.38–2.87 mmol/L (ages 0–2 years), 1.36–2.49 mmol/L (ages 0–1 years) and 163–427 IU/L (ages 0–10 years), respectively.

### 2.4. Biomarkers of Bone Metabolism

The dosage of the analytes (CTX, OC, PINP and Leptin) was obtained, respecting the manufacturer’s indications and carrying out internal quality controls (IQCs).

### 2.5. Collagen Type 1 Carboxyl-Terminal Telopeptide (CTX)

The dosage of CTX was performed using the IDS-ISYS CTX kit based on chemiluminescence technology by the TGSTA Technogenetics analyzer (Technogenetics, Milan, Italy); the luminescence emitted by the labeling with acridinium is directly proportional to the concentration of the analyte present in the analyzed serum sample and is expressed in ng/mL. The limit of detection (LoD) is 0.023 ng/mL, the limit of quantification (LoQ) is 0.033 ng/mL. The analytical coefficient of variation (CVA) (%) ranges from 4.7 to 8.8% depending on the different analyte concentrations in the samples.

### 2.6. Osteocalcin (OC)

Osteocalcin was assayed by chemiluminescence immunoassay using the IDS-iSYS Multi-Discipline Automated System by using the TGSTA Technogenetics analyzer (Technogenetics, Milan, Italy); the luminescence emitted by the labeling with acridinium is directly proportional to the concentration of analyte present in the sample under examination and is expressed in ng/mL. The minimum detectable dose (LoD) is 0.27 ng/mL, the limit of quantification (LoQ) is 1.57 ng/mL. The method has an overall repeatability analytical coefficient of variation (CVA) ranging from 3.7% to 9.2% depending on different analyte concentrations in the samples.

#### Type 1 Amino-Terminal Propeptide (PINP)

PINP was assayed with chemiluminescence technology, using the IDS-iSYS Multi-Discipline Automated System kit by using the TGSTA Technogenetics analyzer (Technogenetics, Milan, Italy); the luminescence emitted by the labeling with acridinium is directly proportional to the concentration of intact PINP in the sample under examination and is expressed in ng/mL; analytical sensitivity limit (LOB) 0.5 ng/mL, the lowest detectable dose (LoD) and limit of quantification (LoQ) are 1.0 ng/mL. The analytical coefficient of variation (CVA) (%) ranges from 4.2 to 5.3% depending on different analyte concentrations in the samples.

### 2.7. Leptin

The Leptin concentration was determined by immunoenzymatic “sandwich” assay (TECO Human Leptin), which involves the use of two mouse monoclonal antibodies with high specificity and affinity for human Leptin. Absorbance reading by the DSX^®^ TGSTA Dynex Technologies, Inc. (Chantilly, VA, USA), and then measurement of the colorimetric reaction, performed at a wavelength of 450 nm, provides the Leptin concentration expressed in ng/mL. According to the manufacturer’s report, the limit of detection (LoD) is 0.25 ng/mL; the limit of quantification (LoQ) is 1 ng/mL; CVA (%) ranges from 7.88% to 19.21%, depending on different analyte concentrations in the samples.

### 2.8. Certification of Analytical Quality

The analytical method imprecision of the bone metabolism analytes included in the study (CTX, OC PINP, Leptin) was determined by calculating the inter-assay (or between day) analytical coefficient f variation (CVA) according to the standard procedure described in document EP15-A3 of the Clinical and Laboratory Standards Institute (CLSI). Internal quality control solutions were used with two levels of concentration (level 1 and level 2) [18]. Each sample was tested in triplicate for five days. The average CVA thus obtained was compared with the analytical quality objectives indicated by the supplier.

### 2.9. Statistical Analysis

The descriptive statistics of the concentrations of biomarkers of bone metabolism assayed in preterm infants included the means, medians and distribution intervals (range and standard deviations-SD). The 95% distribution intervals [confidence intervals (CI)] were calculated using standard parametric and non-parametric statistical analyses. The D’Agostino–Pearson test was used to evaluate the normality of the distribution of the investigated parameters *p* < 0.05. Pearson’s correlation coefficient (r) was used to evaluate the linear correlation between quantitative variables at baseline level (T0) and in the two different sampling times (T1 and T2). The non-parametric Mann–Whitney U test evaluated the differences in the medians of the serum biochemical indicators (Ca, P and ALP) and of the serum biomarkers of bone metabolism (CTX, OC, PINP and Leptin) in the group of subjects stratified by the sampling period (T0, T1 and T2). We used the Wilcoxon test to evaluate the median of the differences between the paired observation of CTX, OC, PINP and Leptin. The difference was statistically significant at *p* < 0.05. Box plots and multiple comparison graphs were used to visualize the distribution of measurements as compared with the time of sampling. The Box plots report the values from the 25th to the 75th quartile, with the central line as the median and the horizontal lines as the extension from the minimum to the maximum value (range) of the concentrations. Multiple comparison graphs allow you to visualize the differences between subgroups (T0, T1, T2,) of the biomarkers of bone metabolism (CTX, OC, PINP and Leptin). The graph is made up of various elements: bars, horizontal lines indicating the extension of the values obtained, connecting lines for the median and dots for identifying all the data. MedCalc software, version 11.6.1.0 (MedCalc Software, Mariakerke, Belgium), was used for statistical analysis.

## 3. Results

### 3.1. Subjects

The mean GA of the neonates was 32.23 ± 2.08 weeks (range 28–35 weeks); mean birth weight 1811 ± 346 g (range 1180–2390 g). Of the 13 preterm neonates, 3 weighed under 1500 g (very low birth weight—VLBW) of which 2 had a history of fetal growth restriction (FGR). In newborns with a GA > 32 weeks, the full enteral feeding was achieved within the first 7 days of life while in those with a GA < 32 weeks, it was achieved within 15 days of life. According to an internal protocol, all preterm neonates with a birth weight under 1700 g received mother’s own milk (MOM) or fortified donor human milk, starting the fortification from an enteral intake of 50 mL/kg/day. Infants with a birth weight >1700 g received MOM or preterm formula. The total PN was prescribed in the first 3 days of life, and in the 3 VLBW neonates who reached full enteral feeding, within 15 days of life.

The mothers of preterm neonates had not received any drugs during pregnancy except for antepartum corticosteroid prophylaxis for RDS. Table 1 shows the clinical characteristics of the study population.

### 3.2. Verification of Analytical Quality

The analytical quality verification of the assay methods using CLSI standard procedure EP15A3 yielded CVAs comparable to those provided by the manufacturer. The CTX, OC, PINP and Leptin assay had an optimal inter-assay CVA (mean CVA: 5.07%). The results are shown in Table 2.

### 3.3. Biochemical Markers

Statistical evaluation of the total and time-stratified (T0, T1, T2) Ca, P and ALP concentrations showed distribution ranges with minimal variations (1% and 3.8%, respectively, for high P limit and low Ca limit) when compared with the reference intervals reported in the literature for the methods used [19,20] (Table 3). The Pearson correlation coefficient summarizes the relationship between the different biochemical variables and highlights a low correlation in the ALP concentrations in the T0 vs. T1 and T2 subgroups and in the Ca concentrations T0 vs. T2 (Table 4). The non-parametric Mann–Whitney U test, used to compare the difference between the medians of the serum biochemical markers (Ca, P and ALP) at T0, T1 and T2, showed a statistically significant difference only for ALP at birth (T0) versus the concentrations at T1 and T2 (Table 4).

The visualization of the Boxplots of Ca, P and ALP confirmed the data obtained from the statistical evaluation. The graphs are shown in Figure 1a–c.

### 3.4. Biomarkers of Bone Metabolism

The descriptive statistics of the distribution of concentrations of the biomarkers with evaluation at the 95% CI and evaluation of the normal distribution (D’Agostino–Pearson test) stratified according to the times of blood sampling (T0, T1, T2) are shown in Table 5.

The Pearson correlation coefficient used to summarize the characteristics of our results and to describe the strength and direction of the linear relationship between the different biomarkers of bone metabolism highlighted an inverse relationship of the trend of Leptin concentration towards CTX and OC and PINP (Table 6). The Mann–Whitney U test showed significant differences between the median concentrations in the T0 group versus the T1 and T2 groups of the biomarkers of bone remodeling, CTX, OC and PINP. A significant difference was also found for the Leptin concentration at time T0 compared to T1.

The median of the differences between the paired observations (Wilcoxon test) were statistically significant (*p* < 0.05) for the concentration of CTX obtained for T0 vs. T1 e T2; of OC T0 vs. OC T1; of P1NP T0 vs. P1NP T2 and of Leptin T0 vs. Leptin T1. The results of the Mann–Whitney U test and Wilcoxon test are shown in Table 7.

The “Multiple Comparison Graphs “ relating to the bone biomarkers Leptin, CTX, OC and PINP report the distribution of analyte concentration measurements according to the sampling time (Figure 2). The graphs show all the values (Dot) in each grouping (T); the bars enclose all the values (range); and the means of the values for each T are connected by a horizontal line. The graphs show a reduction in the median Leptin concentration after birth and an evident increase in the median concentrations of CTX, OC and PINP from T0 to T2. The trend described confirms the differences found with the statistical test comparing the medians.

## 4. Discussion

Fetal bone development is a complex process regulated and supported by minerals, hormones and growth factors transmitted from the mother across the placenta [21,22]. This transfer to the fetus is most evident at the end of pregnancy [23]. Many aspects of this complex process need to be explored further.

This is the first study that evaluated biochemical and bone mineralization markers at birth and within the first month of life in a group of preterm infants who did not present risk factors for MBD. During pregnancy, calcium and phosphate are actively transported from the maternal circulation to the fetus by the placenta. An adequate mineral supplementation during fetal life regulates the activity of osteoblasts, the bone-forming cells, and osteoclasts, the bone-resorbing cells, which are involved in the balanced activity of bone remodeling across the lifespan [23,24]. Premature birth abruptly interrupts the physiological process of the storage and deposition of Ca and P during the period of maximum growth, i.e., the third trimester of pregnancy, and can predispose to MBD [10,24].

The evaluation of serum biochemical markers of bone metabolism such as Ca, P and ALP could be useful for the assessment of mineral deficiency in preterm neonates, although none of them alone can be considered specific for an ongoing MBD [10].

By cutting the umbilical cord and abruptly losing placental Ca infusion, over the course of hours or days, there is a rapid adjustment in the regulation of mineral homeostasis. Birth marks the beginning of a fall in total calcium concentrations, probably caused by the loss of the placental calcium pump, as well as an increase in pH induced by the onset of respiration. These factors contribute to a 20–30% reduction in serum Ca that occurs in the first few hours, followed by an increase in values close to those of adults in the following hours [10]. To maintain a normal blood Ca level during the period of continuous skeletal growth, the neonate becomes dependent on intestinal calcium intake, skeletal and renal calcium stores.

In our study, Ca and P concentrations at birth (time T0) were within the limits of the reference intervals [25].

The assessment of serum Ca levels is not a reliable screening tool because neonates can maintain normal Ca levels despite bone Ca loss. The minimum and maximum total Ca values were 2.23 mmol/L and 2.78 mmol/L, respectively, and the concentrations were within the range of the reference values (2.38–2.87 mmol/L; age 0–2 years) with a greater dispersion of the value to T0 correlated to the different state of prematurity. The comparison between median values and their correlation of median concentrations, at time T0, T1 and T2, did not show any statistically significant differences.

Hypophosphatemia, with serum phosphate levels <1.16 mmol/L (3.6 mg/dL), is the earliest marker of the impaired regulation of mineral metabolism, suggesting mineral depletion and indicating an increased risk of MBD [11,12].

The newborns included in this study showed P concentrations higher than the threshold levels considered at risk for an altered bone development. The 95% CI of the median values ranged from 1.96 mmol/L to 2.18 mmol/L, indicating adequate phosphate levels. The comparison between the median value and the correlation of its concentrations at T0 versus T1 and T2 did not show statistically significant differences in the *p* values.

ALP is considered a bone turnover marker that increases during the first 3 weeks of life, reaching a peak at 6–12 weeks of age (12). ALP levels > 500 IU/L in infants < 30 weeks of gestation have been associated with MBD [10], and values > 700 IU/L suggest bone demineralization, despite the absence of clinical signs [26,27,28].

As reported in the literature, in our group of premature infants, we observed a progressive increase in ALP concentrations from T0 to T1 and T2 [29], with a maximum ALP value recorded (445 IU/L), which was lower than the level considered at risk of developing a bone metabolism disease, suggesting that total ALP could be a confident marker of bone mineral status in preterm infants without a condition predisposing to MBD.

In vitro studies have shown that Leptin stimulates osteoblast differentiation, promotes mineralization and inhibits osteoclastogenesis [28]. In our study, we found that the mean Leptin concentration at birth (T0: 1.85 ng/mL) was higher than the levels obtained at T1 (1.18 ng/mL) and T2 (1.38 ng/mL). These data confirmed the evidence reported in the literature regarding a progressive decrease in Leptin concentrations in the first weeks of life. The rapid decline in circulating Leptin concentrations after birth may be a physiological advantage in preterm infants because it limits body energy expenditure favoring nutritional reserves maintenance that can be used for subsequent growth and development [30,31,32].

As regards the bone metabolism markers, our results show a significant difference in the concentrations of CTX, OC and PINP at different time points, with an increase starting from the first days of life.

The elevation in these markers, consistent with the postnatal profile of biochemical marker levels, may be the result of increased bone remodeling after the deceleration of the growth rate and bone turnover that occurs during the last few weeks of gestation [33]. This increase in CTX, OC and PINP concentrations could also be a consequence of the newborn’s adaptation to extrauterine life conditions (e.g., increased muscle activity, oral nutrition), which induces a significant modulation of bone remodeling.

The concentrations of PINP (95° CI of 3.71 ng/mL and 4.83 ng/mL) found in the observation period were particularly high, especially when compared with those of the adult (95° CI of the mean of apparently healthy subjects between 0.84 and 0.88 ng/mL).

This finding agrees with the results of a previous study in which the authors observed an increase in the markers of bone formation, but not resorption, in VLBW premature infants during the first 10 weeks of life [34].

This can be interpreted on the basis of the different functions that these proteins perform in bone metabolism and in the timing of their release from the bone tissue during the neonatal formation process.

From a clinical point of view, the major problem is the lack of availability of sufficiently sensitive, easily usable and clinically validated biochemical markers for which the variations in concentrations in the first months of life are known.

A recent study has highlighted how only the increase in serum OC could be used as a marker of poor bone mineralization at three months of age [18].

Our approach to assessing the risk of MBD appears to be particularly encouraging and useful since it relied on clinical parameters for the selection of subjects to be included in the study and used a combination of biochemical indicators and biomarkers of bone formation and resorption.

We sought to create a potentially useful reference to evaluate possible deviations in clinical and biochemical measurements in premature infants with conditions at risk of MBD. To this end, the most significant biochemical markers (calcium, phosphate, ALP) and markers of bone metabolism such as PINP, CTx, OC and Leptin were analyzed. The biomarkers were measured using precise analytical methods and with suitable analytical sensitivity.

## 5. Limitations of the Study

The number of subjects included is the limitation of our study. There is an objective difficulty in finding suitable biological material in apparently healthy preterm infants without distinctive signs of MBD of prematurity. Long-term multicenter observational studies, using the same assay method and able to provide suitable analytical performances, could improve the knowledge on metabolic regulatory biomarkers and confirm their utility in the assessment of bone health status in preterm infants.

## 6. Conclusions

In preterm infants with no morbidity and with adequate dietary intake, there is a significant increase in CTX, OC and PINP concentrations during the first few days of neonatal life; the increase is evident in the first days and tends to be less evident subsequently. There is evidence of a reduction in the Leptin concentration after birth.

Different variations in the concentration of these bone remodeling biomarkers in the first weeks of life of preterm infants, integrated by the changes in the biochemical indices (Ca, P, ALP), could be of help in recognizing newborn infants at risk of developing an MBD.

## Figures and Tables

**Figure 1 biomedicines-12-01271-f001:**
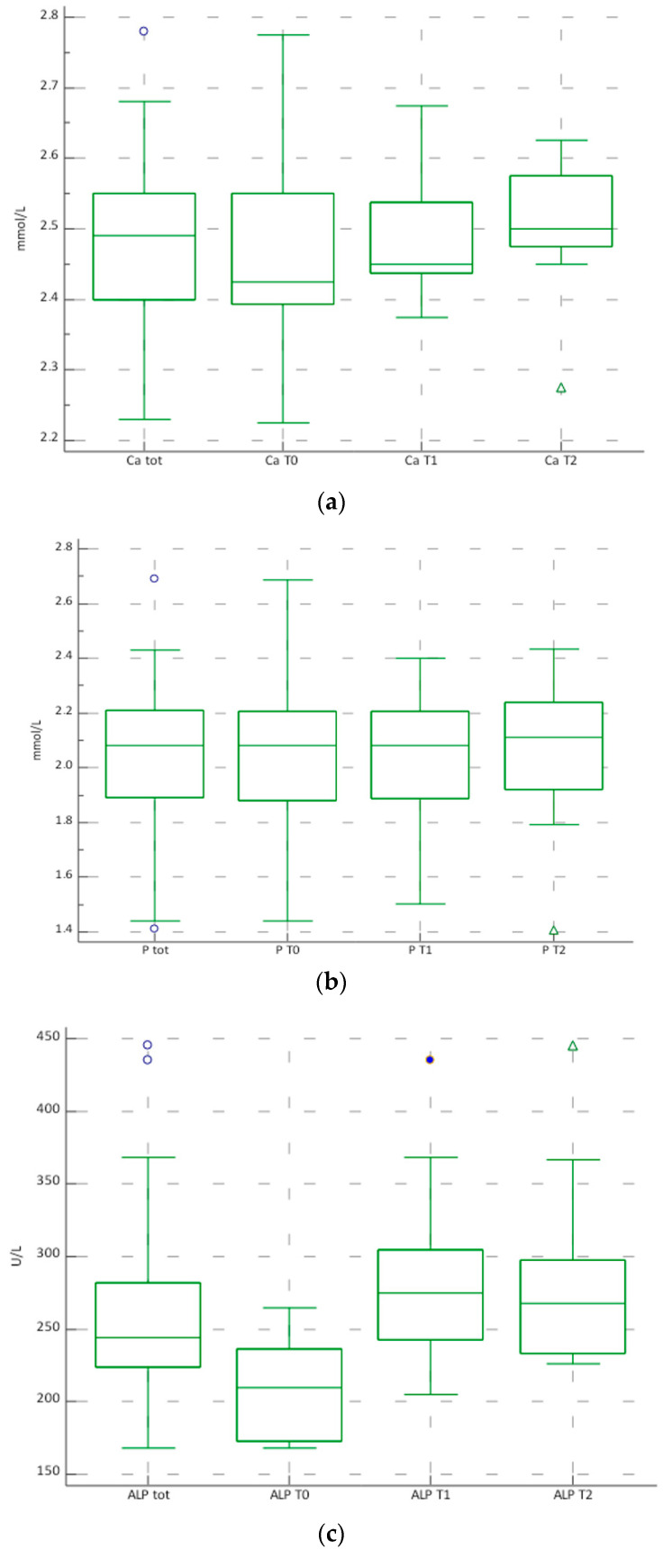
(**a**–**c**). Boxplots of the total concentrations of Ca (mmol/L) (**a**), P (mmol/L) (**b**), ALP (U/L) (**c**) in the subjects evaluated and stratified at time T0, T1, T2.

**Figure 2 biomedicines-12-01271-f002:**
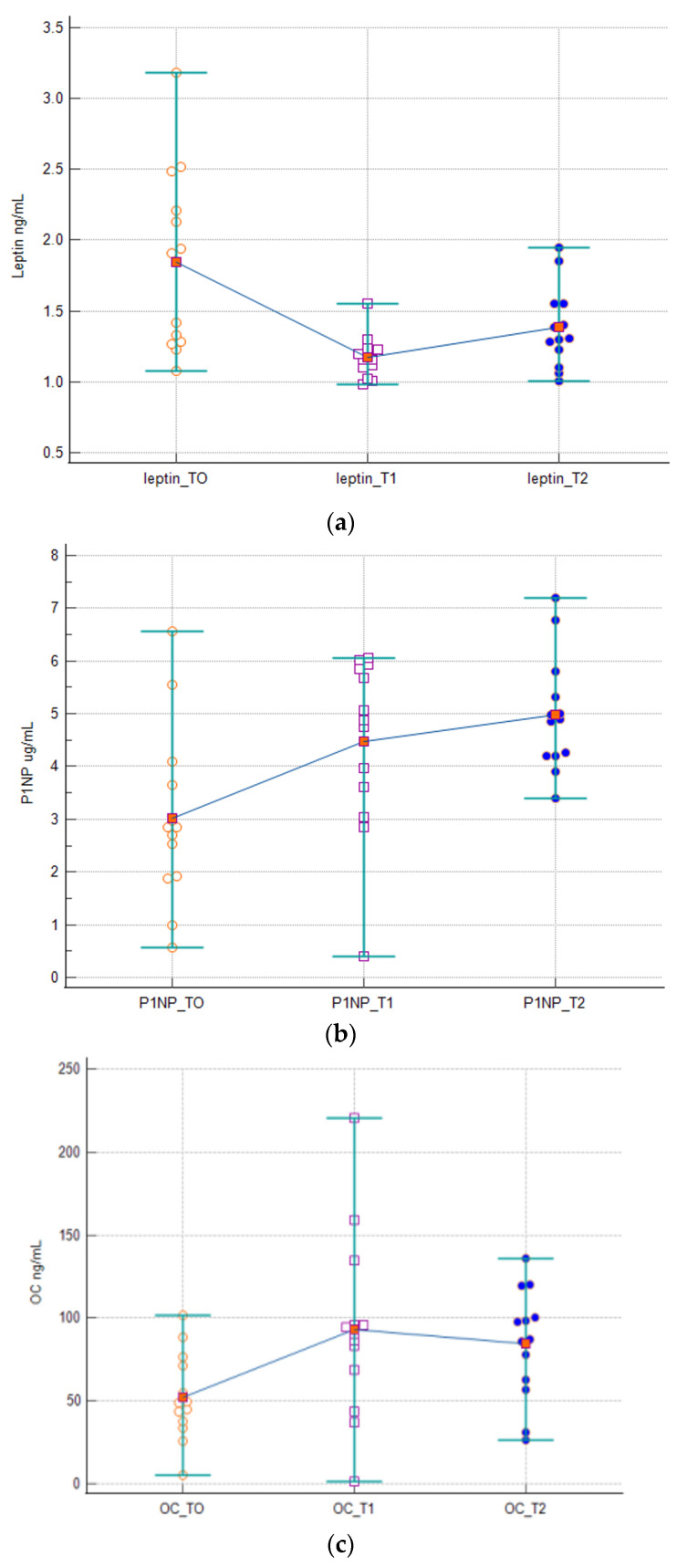
Multiple comparison graphs of the concentrations of Leptin (ng/mL) (**a**), OC (ng/mL) (**b**), PINP (ng/mL) (**c**) and CTX (ng/mL) (**d**) in subjects evaluated and stratified at time T0, T1, T2. The figures represent the dots, the ranges, the median values and the trend line.

**Table 1 biomedicines-12-01271-t001:** Clinical characteristics of the study population.

SEX	GA (Weeks)	BW (g)	BW at 30 Days	BL (cm)	BL at 30 Days	HC (cm)	HC at 30 Days	FG
M	28	1180 (0.52 SDS)	1335 (−1.27 SDS)	36 (−0.27 SDS)	40 (−0.99 SDS)	26 (0.01 SDS)	26.5 (−2.22 SDS)	0
F	29	1350 (0.78 SDS)	1880 (−0.04 SDS)	39 (0.67 SDS)	44.5 (0.59 SDS)	27 (0.35 SDS)	29 (−0.89 SDS)	0
M	30	680 (1.12 SDS)	2170 (−0.17 SDS)	42 (1.08 SDS)	46 (0.34 SDS)	31 (2 SDS)	2 (0.17 SDS)	0
M	32	1400 (−1.07 SDS)	1780 (−2.29 SDS)	39 (−1.39 SDS)	43 (−2.04 SDS)	30 (0.04 SDS)	32 (−0.9 SDS)	1
M	32	2090 (1 SDS)	2530 (−0.55)	43 (0.25 SDS)	46 (−0.79 SDS)	31 (0.79 SDS)	33 (−0.16 SDS)	0
F	32	1680 (−0.01 SDS)	1910 (0.69 SDS)	41 (−0.31 SDS)	43 (0.54 SDS)	30 (0.4 SDS)	32 (1.79 SDS)	0
M	32	2050 (0.88 SDS)	2400 (−0.86 SDS)	43 (0.25 SDS)	46 (−0.79 SDS)	31 (0.72 SDS)	32 (−0.9 SDS)	0
F	33	1630 (−0.76 SDS)	2010 (−2.02 SDS)	40 (−1.29 SDS)	42 (−2.6 SDS)	28 (−1.53 SDS)	29.5 (−2.67 SDS)	0
F	33	2000 (0.28 SDS)	2425 (−1.07 SDS)	43 (−0.06 SDS)	46 (−0.93 SDS)	29 (−0.89 SDS)	31.5 (−0.89 SDS)	0
F	34	2045 (−0.25 SDS)	2400 (−1.61 SDS)	44 (−0.22 SDS)	46 (−1.38 SDS)	32 (0.58 SDS)	33.5 (−0.06 SDS)	0
M	34	1915 (−0.83 SDS)	2200 (−2.36 SDS)	41 (−1.75 SDS)	43 (−2.99 SDS)	31 (−0.52 SDS)	33 (−0.97 SDS)	1
F	35	2170 (−0.55 SDS)	2600 (−1.52 SDS)	46 (0.1 SDS)	48 (−0.76 SDS)	33 (0.78 SDS)	34.5 (0.52 SDS)	0
M	35	2390 (−0.26 SDS)	2800 (−1.39 SDS)	46 (−0.24 SDS)	48.5 (−0.94 SDS)	33 (0.34 SDS)	34.5 (−0.02 SDS)	0

M: male; F: female; GA: gestational age; BW: birth weight; BL: birth length; HC: head circumference; FGR: fetal growth restriction.

**Table 2 biomedicines-12-01271-t002:** The inter-assay analytical coefficient of variation (CVA) obtained according to the standard EP15-A3 using the internal quality control (IQC) material.

Variables	CTX (ng/mL)		OC (ng/mL)		PINP (ng/mL)		Leptin (ng/mL)	
IQC	Level 1	Level 2	Level 1	Level 2	Level 1	Level 2	Level 1	Level 2
Mean	0.80	1.98	7.10	20.89	20.59	46.44	8.81	28.61
SD	0.04	0.09	0.29	0.92	1.01	2.50	0.62	1.31
CVA inter-assay	5.30	4.71	4.18	4.42	4.94	5.39	7.05	4.59

Analytical quality verification of assay methods using CLSI standard procedure EP15A3 yielded CVAs comparable to those provided by the manufacturer. CTX, OC, PINP and Leptin assay had optimal inter-assay CVA (mean CVA: 5.07%).

**Table 3 biomedicines-12-01271-t003:** Descriptive statistics of Ca, P and ALP concentrations in the subjects evaluated (total) and stratified at times T0, T1, T2.

	Ca (mmol/L)				P (mmol/L)				ALP (UI/L)			
	Total	T0	T1	T2	Total	T0	T1	T2	Total	T0	T1	T2
N	39	13	13	13	39	13	13	13	39	13	13	13
Min	2.28	2.28	2.38	2.45	1.41	1.44	1.50	1.41	168.0	168.0	205.0	226.0
Max	2.78	2.78	2.68	2.63	2.63	2.63	2.40	2.43	440.0	265.0	435.0	440.0
Mean	2.49	2.48	2.49	2.50	2.05	2.04	2.05	2.05	256.0	208.76	282.84	285.80
95% CI	2.45 to 2.53	2.39 to 2.56	2.43 to 2.54	2.43 to 2.57	1.95 to 2.14	1.85 to 2.23	1.90 to 2.20	1.84 to 2.25	234.0 to 279.0	187.74 to 229.79	245.41 to 320.27	234.87 to 336.73
Median	2.49	2.43	2.45	2.50	2.08	2.08	2.08	2.11	244.0	210.0	275.0	267.50
95% CI	2.45 to 2.51	2.39 to 2.55	2.42 to 2.55	2.46 to 2.59	1.96 to 2.18	1.87 to 2.21	1.89 to 2.23	1.85 to 2.27	230.0 to 268.0	172.12 to 237.74	240.37 to 308.02	231.42 to 334.22
Normal distribution	*p* = 0.568	*p* = 0.501	*p* = 0.308	*p* = 0.0547	*p* = 0.4197	*p* = 0.7046	*p* = 0.4387	*p* = 0.1081	*p* = 0.0027	*p* = 0.2590	*p* = 0.0256	*p* = 0.0432

N: number of samples analyzed; CI: confidence interval. Statistical evaluation of the total and time-stratified (T0, T1, T2) Ca, P and ALP concentrations showed distribution ranges with minimal variations when compared with reference intervals reported in the literature for the methods used.

**Table 4 biomedicines-12-01271-t004:** Comparison between medians (Mann–Whitney U test) and correlation (Pearson) between biochemical markers, stratified at T0, T1, T2.

Variables	Mann–Whitney U Test	Pearson Correlation (r)
Ca T0 vs. Ca T1	*p* = 0.5877	0.47
Ca T0 vs. Ca T2	*p* = 0.4168	0.03
Ca T1 vs. Ca T2	*p* = 0.3633	0.31
P T0 vs. P T1	*p* = 0.7328	0.47
P T0 vs. P T2	*p* = 0.7328	0.46
P T1 vs. P T2	*p* = 0.8522	0.78
ALP T0 vs. ALP T1	*p* = 0.0010	0.04
ALP T0 vs. ALP T2	*p* = 0.0035	0.07
ALP T1 vs. ALP T2	*p* = 0.8523	0.73

The Pearson correlation coefficient summarizes the relationship between the different biochemical variables and highlights a low correlation in the ALP concentrations in the T0 vs. T1 and T2 subgroups and in the Ca concentrations T0 vs. T2. The non-parametric Mann–Whitney U test, used to compare the difference between the medians of the serum biochemical markers (Ca, P and ALP) at T0, T1 and T2, showed a statistically significant difference only for ALP at birth (T0) versus concentrations at T1 and T2.

**Table 5 biomedicines-12-01271-t005:** Descriptive statistics of CTX, OC, PINP and Leptin concentrations in the subjects evaluated (total) and stratified at times T0, T1, T2.

	CTX (ng/mL)				OC (ng/mL)				PINP (ng/mL)				Leptin (ng/mL)			
	Total	T0	T1	T2	Total	T0	T1	T2	Total	T0	T1	T2	Total	T0	T1	T2
**N**	39	13	13	13	39	13	13	13	39	13	13	13	39	13	13	13
**Min**	0.25	0.25	0.30	0.39	2.1	5.60	2.1	26.10	0.50	0.58	0.50	3.40	1.01	1.08	0.98	1.01
**Max**	1.48	0.47	1.25	1.48	220	101.4	220.0	135.80	7.19	6.56	6.06	7.19	3.18	3.18	1.55	1.95
**Mean**	0.63	0.34	0.77	0.80	75.86	52.35	93.09	83.00	4.27	3.01	4.47	5.06	1.48	1.85	1.18	1.38
**95% CI**	0.52 to 0.74	0.29 to 0.38	0.58 to 0.95	0.59 to 1.01	61.17 to 90.54	36.51 to 68.19	59.58 to 126.6	58.72 to 107.27	3.71 to 4.83	2.00 to 4.02	3.46 to 5.47	4.28 to 5.85	1.31 to 1.65	1.46 to 2.23	1.09 to 1.27	1.17 to 1.58
**Median**	0.53	0.32	0.75	0.78	73.8	48.90	90.60	85.80	4.23	2.85	4.87	4.98	1.28	1.91	1.16	1.31
**95% CI**	0.42 to 0.76	0.29 to 0.41	0.54 to 1.02	0.54 to 0.97	49.16 to 89.02	35.57 to 73.65	56.88 to 114.0	52.02 to 119.28	3.54 to 5.15	1.90 to 3.86	3.35 to 5.89	4.15 to 5.98	1.23 to 1.41	1.27 to 2.34	1.06 to 1.23	1.09 to 1.60
**Normal** **distribution**	*p* = 0.0047	*p* = 0.11	*p* = 0.65	*p* = 0.48	*p* = 0.0222	*p* = 0.93	*p* = 0.42	*p* = 0.74	*p* = 0.5608	*p* = 0.45	*p* = 0.05	*p* = 0.66	*p* = <0.0001	*p* = 0.20	*p* = 0.18	*p* = 0.27

N—number of samples analyzed; CI—confidence interval. The descriptive statistics of the distribution of concentrations of the biomarkers with evaluation at the 95% CI and evaluation of the normal distribution (D’Agostino–Pearson test) stratified according to the times of blood sampling (T0, T1, T2).

**Table 6 biomedicines-12-01271-t006:** Correlation (Pearson) between biomarkers of bone metabolism (CTX, PINP, OC and Leptin).

Correlation between Variables	Total CTX vs. Total Leptin	Total CTX vs. Total OC	Total CTX vs. Total PINP	Total Leptin vs. Total OC	Total Leptin vs. Total PINP	Total OC vs. Total PINP
Correlation coefficient (r)	−0.27	0.64	0.46	−0.23	−0.43	0.44
P	0.1037	<0.0001	0.0042	0.1780	0.0082	0.0069

The Pearson correlation coefficient highlighted an inverse relationship of the trend of Leptin concentration towards CTX and OC and PINP.

**Table 7 biomedicines-12-01271-t007:** Comparison of medians (Mann–Whitney U test and Wilcoxon test) of bone metabolism biomarkers stratified at T0, T1, T2.

Variables	Mann–Whitney U Test (Independent Samples)	Wilcoxon Test (Paired Samples)
	Two-Tailed Probability	Two-Tailed Probability
Leptin T0 vs. Leptin T1	*p* = 0.0009	*p* = 0.0098
Leptin T0 vs. Leptin T2	*p* = 0.0679	*p* = 0.0840
Leptin T1 vs. Leptin T2	*p* = 0.0677	*p* = 0.1055
CTX T0 vs. CTX T1	*p* = 0.0002	*p* = 0.0039
CTX T0 vs. CTX T2	*p* = 0.0001	*p* = 0.0020
CTX T1 vs. CTX T2	*p* = 0.8167	*p* = 0.1055
OC T0 vs. OC T1	*p* = 0.0333	*p* = 0.0445
OC T0 vs. OC T2	*p* = 0.0397	*p* = 0.0840
OC T1 vs. OC T2	*p* = 0.8393	*p* = 0.9219
PINP T0 vs. PINP T1	*p* = 0.0293	*p* = 0.1055
PINP T0 vs. PINP T2	*p* = 0.0028	*p* = 0.0098
PINP T1 vs. PINP T2	*p* = 0.6224	*p* = 0.6250

Mann–Whitney U test showed significant differences between the median concentrations in the T0 group versus the T1 and T2 groups of the biomarkers of bone remodeling, CTX, OC and PINP. A significant difference was also found for Leptin concentration at time T0 compared to T1. Wilcoxon test showed significant statistically difference (*p* < 0.05) for the concentration of CTX obtained for T0 vs. T1 e T2; of OC T0 vs. OC T1; of P1NP.

## Data Availability

Data are contained within the article.

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
