# Peer review of "Biomarkers and Biochemical Indicators to Evaluate Bone Metabolism in Preterm Neonates"

_biomedicines, 2024, doi:10.3390/biomedicines12061271_

Round 1

Reviewer 1 Report

Comments and Suggestions for Authors

Dear Authors,

Thank you for the opportunity to review this article. The authors presented interesting results regarding the importance of bone remodeling biomarkers in the first weeks of life of premature infants and their connections with biochemical parameters (Ca, P, Alp). The results may be highly useful in diagnosing the biological condition of bones in premature infants.

Personally, I consider detailed descriptions of methods for performing biochemical analyzes of each parameter to be unnecessary.

In my opinion, it is highly advisable to show the biological and social background of the analyzed newborns. What was the biological age of their mothers, whether they were primiparous or multiparous, what was their BMI, education, degree of urbanization of the place of origin, etc. Many studies indicate the high importance of epigenetic (and of course genetic) factors in shaping the biological condition of offspring. If the authors have such data, please provide it. If they are missing, this should be mentioned in the limitation study.

Kind regards,

reviewer

Author Response

Thanks for the review provided.
In replay to: "
Personally, I consider detailed descriptions of methods for performing biochemical
analyzes of each parameter to be unnecessary." The authors believe that the detailed description of the methods for the measurement of biomarkers of bone metabolism is an essential
requirement to provide evidence that the data provided for the single analyte responded to sensitivity (LoD and LoQ) and specificity (immunometric)
criteria suitable for the study

In replay to: "In my opinion, it is highly advisable to show the biological and social background of the analyzed newborns. What was the biological age of their mothers, whether they were primiparous or multiparous, what was their BMI, education, degree of urbanization of the place of origin, etc. Many studies indicate the high importance of epigenetic (and of course genetic) factors in shaping the biological condition of offspring. If the authors have such data, please provide it. If they are missing, this should be mentioned in the limitation study.

The authors have integrated the available data into the materials and methods part.
As regards micro RNAs and the use of instrumental technologies such as REMS,
although they provide useful information in assessing the fetal risk of developing osteopenia, they are not
used in clinical routine

Reviewer 2 Report

Comments and Suggestions for Authors

Dear Author, 

Thank you for the opportunity to review this article.

The Abstract is relevant.

The introduction lacks a proper structure and should be more punctuated on knowledge on each marker and serum level and the general implications on principal bone diseases. Elaboration on each pathology should be noted in the Discussion. Please rephrase.

Materials and Results are well made. The patient pool is rather small. Is the distribution normal?

Conclusions need to comprise 3-7 clear phrases based on your results. Please reformulate

Author Response

Thanks for the review provided

In replay to: The introduction lacks a proper structure and should be more punctuated on knowledge on each marker and serum level and the general implications on principal bone diseases. Elaboration on each pathology should be noted in the Discussion. Please rephrase.

The authors improved the introduction and provided the following information:
An increase in bone turnover has been previously reported in preterm infants,
especially those with MBD. Several authors have highlighted that
an approach to the diagnosis of MBD based on a combination of different
biomarkers of bone formation and resorption appears to be more useful.
In fact, the indices of bone formation (OC, PINP) and resorption (CTx)
are significantly increased in the group of newborns with MBD,
both term and premature. In particular, OC appears to be a very
specific predictor of reduced bone mass in premature infants with MBD".

In replay to: "Materials and Results are well made. The patient pool is rather small."

The authors thank the comment, unfortunately they were forced to adequately
select the sample, for this reason the number of those enrolled was limited.

in replay:" Is the distribution normal? "

The authors state that not all parameters considered had a normal distribution and to make this answer clear they modified tables 3 and 4 by inserting the P value.

In replay to: "Conclusions need to comprise 3-7 clear phrases based on your results. Please reformulate"

The authors reformulated the conclusions

Round 2

Reviewer 2 Report

Comments and Suggestions for Authors

The paper is ready to be publish